# DSTN: Early Spatio-Temporal Forecasting with Dynamic Propagation

## Abstract

In most spatio-temporal prediction tasks, the timeliness of predictions is more critical than their accuracy. For instance, in tasks such as crime prediction, traffic congestion forecasting, and wildfire early warning, waiting longer to gather additional information may improve prediction accuracy, but it does not provide enough preparation time for subsequent actions, rendering the precise predictions valueless. Therefore, balancing between prediction timeliness and accuracy is essential for such tasks. In this paper, we propose an adaptive early spatio-temporal prediction model with a dynamic propagation matrix (DSTN), which captures causal relationships between nodes to enhance prediction timeliness while maintaining accuracy. Our model makes the following contributions: (1) Exploiting the similar long-term patterns of node signals for early prediction. (2) Proposing the concept of Asynchronous Spatio-temporal Causal Frame Pair to effectively capture the spatio-temporal causal relationships between different nodes. (3) Constructing a dynamic propagation matrix to filter out irrelevant information for early prediction. Experimental results on four large-scale real-world datasets demonstrate that the performance of our proposed DSTN model generally outperforms all baselines. The source code is available at `https://anonymous.4open.science/r/DSTN-DB49`.

## 1 Introduction

Spatio-temporal prediction is essential for understanding and forecasting the dynamics of complex systems that evolve across both time and space, making it a powerful tool in a wide range of application such as traffic planning, environmental monitoring, and disease control (Pan et al., 2019; Catlett et al., 2019; Hengl et al., 2012; Gao et al., 2021; Cheng & Wang, 2008).

Traditional spatio-temporal prediction methods primarily focus on improving prediction accuracy while neglecting timeliness (when to make predictions). However, in most cases, the timeliness of spatio-temporal predictions is more important than accuracy. For instance, in crime prediction, we prefer to make relatively reliable predictions as early as possible rather than waiting for more accurate predictions at a later time. Early crime prediction results allow police officers to better plan patrol routes or proactively deploy personnel in high-risk crime areas. Earlier predictions not only optimizes resource allocation but also increases the probability of successfully preventing crimes or apprehending suspects.

Existing early prediction methods can be divided into fixed early prediction and adaptive early prediction. Figure 1 compares the early prediction approaches with traditional methods. The traditional method provides prediction results after the complete observation window (Graves & Graves, 2012; Wen et al., 2019). This method is not suitable for tasks that are sensitive to prediction timing, as it does not make any predictions until the last moment $t_p = T$. In fixed early prediction methods, the predictions are made at a fixed time $t_p < T$ (Li et al., 2017). This method manually advances the prediction time $t_p$ without waiting for the complete sequential data to be observed, which improves prediction timeliness to some extent. However, a fixed prediction time cannot provide an optimal prediction for all nodes.

While a larger observation window is intuitively expected to improve prediction accuracy, it often contains redundant information that fails to enhance both accuracy and timeliness. For signals with stable patterns, equivalent or better predictions can be achieved using minimal information, without

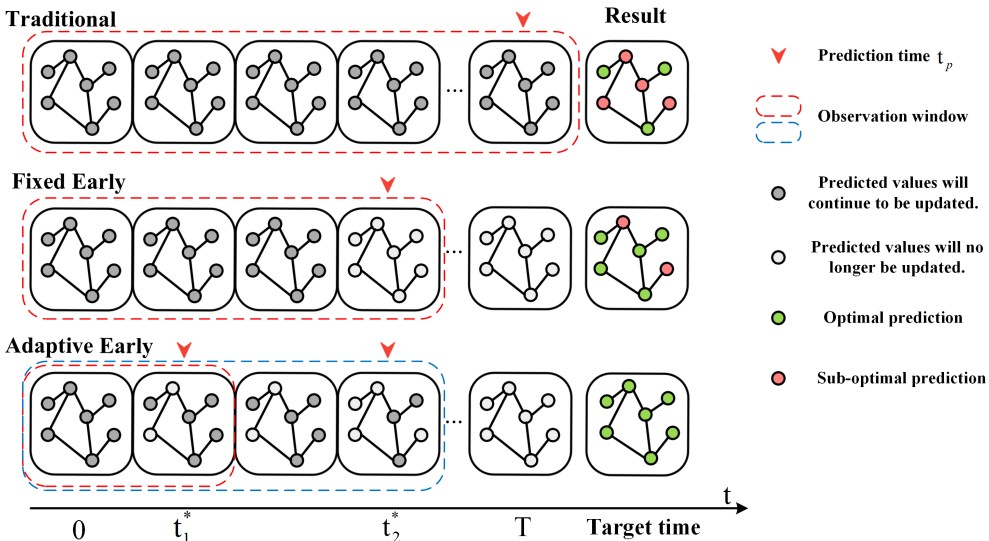

Figure 1: Examples of traditional and early prediction methods. In the adaptive early prediction, there are multiple prediction time at which one or more nodes stop updating their predicted values for the target time.

waiting for the full window to conclude. Therefore, for each node, there exists an optimal prediction time based on the current signal dynamics, and fixed early prediction methods fail to align with the optimal timing. In adaptive early prediction, each node makes predictions at its own optimal prediction time, determined by a module within the early prediction model. At each time step, if a node reaches this point, its predicted value is output. As the data dynamically evolves, the optimal prediction time for each node will also fluctuate. For example, in the Figure 1, at time points $t_1^*$ and $t_2^*$, some nodes reach their optimal prediction time and make their final predictions. These predicted values for the target time will not be updated in subsequent time steps.

Although adaptive early prediction has been initially explored in (Shao et al., 2024; 2023), establishing a framework based on dual-objective reinforcement learning, several issues remain: (1) **Ignoring the long-term periodic characteristics of signals**: The long-term pattern of signals is an important feature that can be leveraged in early spatio-temporal prediction. Nodes with highly consistent long-term patterns are likely to have direct correlations. However, existing methods primarily focus on short-term dynamic patterns and overlook these long-term periodic characteristics. (2) **Failure to capture the causality of signal dynamics**: Capturing the causal relationship can effectively utilize the historical observations of causal nodes to guide the early prediction of the target node. Dynamics between signals often exhibit strong causal relationships, which is particularly evident in scenarios such as traffic flow prediction and disease outbreak forecasting. For example, in traffic flow prediction tasks, the traffic volume at the target node relies on that at adjacent nodes in nearby time steps, exhibiting a strong causal relationship. (3) **No filtering of irrelevant information**: For the target node, only information relevant to its current dynamics can assist in making an early prediction, while extraneous information may disrupt the model's ability to correctly interpret the target node's current pattern. For example, consider two traffic nodes $u$ and $v$, where $u$ is upstream of $v$ with an average travel time of one hour. For predicting traffic at node $v$ at time $t$, the most relevant input is the traffic at node $u$ at time $t-1$, while traffic from earlier time steps (e.g., $t-3$) is irrelevant. Existing models often fail to filter out such irrelevant signals, leading to degraded performance.

To address the above issues, we propose an adaptive early spatio-temporal prediction model with a dynamic propagation matrix (DSTN). The specific contributions of this paper can be summarized as follows:

- We propose an adaptive early spatio-temporal prediction model with a dynamic propagation matrix, which can effectively balance the timeliness and accuracy of predictions in various spatio-temporal prediction scenarios.

- We compute a frequency similarity matrix to measure the correlations of long-term patterns between nodes, providing essential information for early prediction.

- We propose the concept of Asynchronous Spatio-temporal Causal Frame Pair for the first time, which effectively captures the short-term causal relationships of node dynamics across temporal and spatial dimensions.

- We constructed a comprehensive dynamic propagation matrix to filter out irrelevant information for early spatio-temporal prediction.

## 2 RELATED WORKS

### 2.1 SPATIO-TEMPORAL FORECASTING

Spatio-temporal prediction is clearly crucial in a wide range of real-world applications (Atluri et al., 2018), such as traffic prediction (Lana et al., 2018), crime prediction (Zhao & Tang, 2017), and disease prediction (Waller et al., 1997). Early approaches employed traditional time-series methods such as ARIMA (Pan et al., 2012) and ES (Williams et al., 1998), but they are unable to make use of spatio-temporal dependencies inherent to the data. With the development of Recurrent Neural Networks, deep learning methods such as Long Short Term Memory (LSTM) (Gupta et al., 2023) and Gate Recurrent Unit (GRU) (Pan et al., 2019) are extensively used for capturing temporal patterns embedded in the data. Due to the graph structure of spatio-temporal data, Graph Convolution Networks (GCNs) (Kipf & Welling, 2016; Lu et al., 2020; Dong et al., 2024) are utilized to exploit spatial correlations in non-Euclidean space. Despite the effectiveness of these methods, flexible forecasting can not be realized because of the fixed observation windows. When dealing with rapidly changing data sources, an adaptive forecasting strategy is necessary to ensure the timeliness and accuracy of the prediction.

### 2.2 EARLY PREDICTION

Early Prediction of spatio-temporal data is especially important in some time sensitive applications such as epidemic forecasting. It aims to forecast the incoming events as soon as possible, while maintaining acceptable levels of accuracy in the predictions. Shapelets proposed in (Ye & Keogh, 2009) are sub-parts of time series that represent a class as much as possible. It offers an effective and interpretable method for early prediction (Xing et al., 2011; Ghalwash & Obradovic, 2012; He et al., 2015). Another solution is to build a set of predictors with one or more rules or trigger functions to evaluate the reliability of predictions (Mori et al., 2017a;b; Shao et al., 2024). At each time step, the predictors determine whether the predictions should be preserved or discarded. Although these methods offer an option for adjusting the trade-off between accuracy and timeliness, it is realized by parameters set in advance, and the parameters cannot adjust automatically with evolving data patterns.

In this paper, we propose an adaptive early spatio-temporal prediction model with a dynamic propagation matrix, which improves the timeliness and accuracy of early predictions by leveraging the patterns and causality of node signals, addressing several issues in existing early spatio-temporal prediction methods.

## 3 PROBLEM DEFINITION

Given a graph $G = (\mathbb{V}, \mathbb{E})$, where $\mathbb{V} = \{v_i\}_{i=1}^n$ represents the set of nodes, and $\mathbb{E} = \{e_{ij}\}_{i,j=1}^n$ represents the set of edges. The adjacency matrix $\boldsymbol{A} \in \mathbb{R}^{n \times n}$ captures the geographic relationships between nodes. The observed values $X_t = \{x_t^i\}_{i=1}^n$ refer to the measurements obtained from the nodes at time $t$. The predicted values $\hat{X}_T = \{\hat{x}_T^i\}_{i=1}^n$ represent the estimated values at time $T$. For instance, in a traffic speed forecasting scenario, $x_t^i$ could denote the speed at sensor $v_i$ at time $t$, and $\hat{x}_T^i$ would be the predicted speed at time $T$. The optimal prediction time series $t^* = \{t_i^*\}_{i=1}^n$ denote the time that maximizes prediction accuracy while minimizing time cost for each node. The objective is to determine the optimal prediction time under the current balance between prediction accuracy and timeliness. The optimal prediction time for node $v_i$ is computed with the following

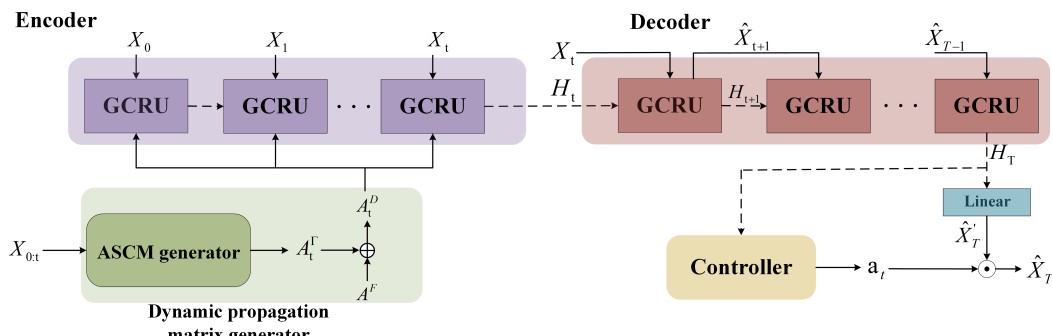

Figure 2: The framework of DSTN.

function:

$$t_i^* = \arg\max_t \left( \log P \left( \hat{x}_T^i \mid X_{0:t}, G \right) - \text{Cost}(t) \right) \tag{1}$$

where $t = 0, 1, ..., T-1$, $T$ is the length of the complete observation window that used for prediction. The part Cost(t) represents the time cost of the prediction, and it is assumed to be a monotonically increasing function about $t$.

## 4 METHOD

As illustrated in Figure 2, the adaptive early spatio-temporal prediction model with a dynamic propagation matrix (DSTN) comprises three key components: a dynamic propagation matrix generator, a spatio-temporal encoder-decoder pair, and a controller for determining the optimal prediction time. The detailed formulation of the loss function is provided in the Appendix A.

### 4.1 DYNAMIC PROPAGATION MATRIX GENERATOR

Figure 3 shows a pair of node signals with similar long-term periodic patterns and short-term dynamic causal relationship. The following insights can be drawn from the figure: (1) Even nodes with similar long-term patterns may only show causal connections during specific short-term dynamic periods—as seen between nodes 43 and 93 from 16:00 to 20:00—while remaining unrelated at other times, such as around 15:00. (2) Measuring causality solely based on the shape of signals over the entire observation window is inefficient and fails to capture localized and temporally relevant causal interactions. (3) A time delay exists due to the physical propagation of dynamics between nodes; this delay helps extract the relevant signal frames and identify predictive segments in already-observed nodes that indicate impending behavior in the target node, supporting early prediction.

To fully leverage the aforementioned data patterns, we construct a dynamic propagation matrix. First, we use the Discrete Fourier Transform (DFT) to convert the signals into frequency domain, where we filter out nodes with similar long-term periodic patterns based on their amplitude-frequency characteristics. The phase difference between signals is then used to estimate the time delay. Using the time delay, we extract the Asynchronous Spatio-temporal Causal Frame Pairs, consisting of a posterior frame and a prior frame. Then the Dynamic Time Warping (DTW) distance between the frame pairs is calculated to measure the causal relationship between nodes regarding current dynamics. Finally, a dynamic propagation matrix is constructed based on theses factors to propagate the corresponding early prediction information to the target node.

For each node $v_i$ in the graph, the DFT is first applied to the historical observations $s_i^{t_H} \in \mathbb{R}^{t_H}$. The length of the time series $t_H$ is set according to the specific periodic characteristics of the data, which can correspond to data from one day, one week, or one month. We obtain the corresponding frequency-domain information $\mathcal{F}_i \in \mathbb{R}^{t_H/2}$:

$$\mathcal{F}_i = |DFT(s_i^{t_H})| \tag{2}$$

Note that we only consider the positive frequency part of the amplitude-frequency spectrum, so the dimension of $\mathcal{F}_i$ is $t_H/2$. The vector $\mathcal{F}_i$ contains the amplitude-frequency information of the node,

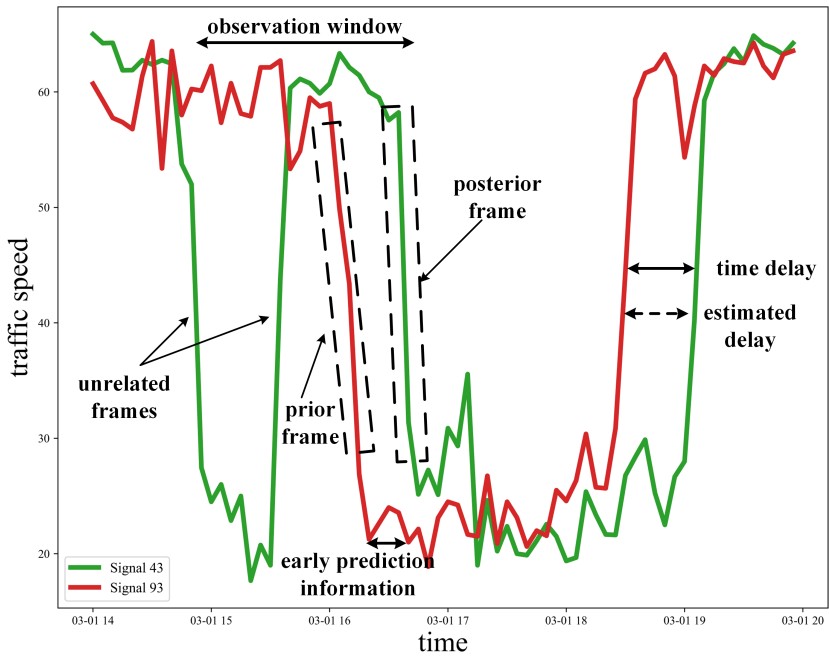

Figure 3: An example of signal segments from a pair of nodes with similar long-term temporal patterns. Asynchronous spatio-temporal causal frame pairs are located within the observation window by estimating the time delay and transmit the corresponding early prediction information.

with a dominant frequency $f^i$, indicating that the energy of node $v_i$ is maximized at frequency $f^i$. The frequency similarity matrix $\boldsymbol{A}^F \in \mathbb{R}^{n \times n}$ is calculated based on the cosine similarity of the amplitude-frequency vectors of the nodes:

$$\boldsymbol{A}_{i,j}^F = \frac{\mathcal{F}_i \cdot \mathcal{F}_j}{\|\mathcal{F}_i\|\|\mathcal{F}_j\|} \tag{3}$$

From the DFT results of the node signals, the phase angle at the dominant frequency can be obtained. For signals with the same dominant frequency, the time delay $t_\Delta$ between node signals can be estimated based on their dominant frequency and phase shift. The equation is as follows:

$$t_\Delta^{ij} = \begin{cases} \dfrac{\Delta\theta_{ij}}{2\pi f^i}, & f^i = f^j \\ 0, & f^i \neq f^j \end{cases} \tag{4}$$

where $\Delta\theta_{ij} = \theta(f^i) - \theta(f^j)$, and $\theta(f)$ means the phase of $f$. As shown in the Figure 3, the estimated time delay (approximately 28 minutes) is close to the actual time delay (approximately 30 minutes). Then, the Asynchronous Spatio-temporal Causal Matrix (ASCM) $\boldsymbol{A}_t^{\mathcal{T}} \in \mathbb{R}^{n \times n}$ is computed using the following formula:

$$\boldsymbol{A}_{t,i,j}^{\mathcal{T}} = \begin{cases} e^{-\alpha \times d_{i,j}}, & i \neq j \ and \ t_\Delta^{ij} < 0 \\ 0, & i = j \ or \ t_\Delta^{ij} \geq 0 \end{cases} \tag{5}$$

$$d_{i,j} = DTW(\Delta x_{(t+t_\Delta^{ij}):t}^i, \Delta x_{(t+2t_\Delta^{ij}):(t+t_\Delta^{ij})}^j) \tag{6}$$

where $\Delta x_{0:t}^i = x_{1:t}^i - x_{0:t-1}^i$, and $\alpha$ scales the DTW distance to an appropriate range, which can be adjusted according to the actual characteristics of the data. Performing a first-order difference on the time series helps eliminate the impact of absolute amplitude on the DTW-based similarity calculation. By incorporating the time delay between signals, we extract the asynchronous spatio-temporal causal frame pair: posterior frame $\Delta x_{(t+t_\Delta^{ij}):t}^i$ and prior frame $\Delta x_{(t+2t_\Delta^{ij}):(t+t_\Delta^{ij})}^j$ to calculate the

node causality in the current time period. The condition $t_\Delta^{ij} < 0$ implies that the change in node $v_j$ occurs prior to the change in node $v_i$.

To accurately propagate the early prediction information in the signal after the prior frame, we need to apply a masked operation to ASCM based on the time delay to filter out irrelevant information. The propagation matrix $\boldsymbol{A}_{t,t'}^{\mathcal{T}} \in \mathbb{R}^{n \times n}$ for each retrospective time step $t' \leq t$ is:

$$\boldsymbol{A}_{t,t',i,j}^{\mathcal{T}} = \begin{cases} \boldsymbol{A}_{t,i,j}^{\mathcal{T}}, & t' \geq t + t_\Delta^{ij} \\ 0, & t' < t + t_\Delta^{ij} \end{cases} \tag{7}$$

The final dynamic propagation matrix $\boldsymbol{A}_{t,t'}^{\mathcal{D}} \in \mathbb{R}^{n \times n}$ is constructed by combining $\boldsymbol{A}^F$ and $\boldsymbol{A}_{t,t'}^{\mathcal{T}}$:

$$\boldsymbol{A}_{t,t'}^{\mathcal{D}} = \boldsymbol{A}^F + \boldsymbol{A}_{t,t'}^{\mathcal{T}} \tag{8}$$

## 4.2 SPATIO-TEMPORAL ENCODE-DECODER PAIR

To fully leverage the early prediction benefits brought by the dynamic propagation matrix, we use Graph Convolutional Recurrent Unit (GCRU) as the fundamental unit for both the encoder and decoder. By incorporating graph convolutions within recurrent cells, GCRU is able to effectively capture both the temporal dynamics and the spatial relationships present in the data. The computation of GCRU can be represented as $GCRU(X, \boldsymbol{A})$, where $X$ is the input graph signal and $\boldsymbol{A}$ is the propagation matrix used for graph convolution.

At each time step $t$, the recorded values of graph signals $X_{0:t}$ are used as the input to encoder GCRU units, with the initial hidden state being the output of the encoder at time $t-1$. For $t' <= t$, the calculation formula of the encoder is:

$$Encoder(X_{t'}) = \begin{cases} GCRU(X_{t'}, \boldsymbol{A}_{t,t'}^{\mathcal{D}}), & t' < t \\ GCRU(X_{t'}, \boldsymbol{A}_{t,t'}^{\mathcal{D}} + \hat{\boldsymbol{A}}^S), & t' = t \end{cases} \tag{9}$$

where $\hat{\boldsymbol{A}}^S \in \mathbb{R}^{n \times n}$ is the normalized predefined distance adjacency matrix $\boldsymbol{A}^S \in \mathbb{R}^{n \times n}$: $\hat{\boldsymbol{A}}^S = \tilde{\boldsymbol{D}}^{\frac{1}{2}} \tilde{\boldsymbol{A}}^S \tilde{\boldsymbol{D}}^{-\frac{1}{2}}$, $\tilde{\boldsymbol{A}}^S = \boldsymbol{A}^S + \boldsymbol{I}$, $\boldsymbol{I}$ is the identity matrix, and $\tilde{\boldsymbol{D}} = \sum_j \tilde{\boldsymbol{A}}_{i,j}^S$.

The final output of the encoder $H_t$ serves as the initial hidden state for the decoder GCRU unit, with the last observation $X_t$ acting as the initial input. Subsequently, the predicted values $\hat{X}_{t+1}$ are used as inputs for the next decoder GCRU unit. In the decoder, a new asynchronous spatio-temporal causal matrix will not be calculated; instead, the causal matrix at time $t$ from the encoder is reused for the GCRU computations. The output of the last decoder unit $H_T$, after passing through a fully connected layer, yields the candidate prediction results $\hat{X}_T'$. The prediction values $\hat{X}_T'$ will be computed with the actions $\mathbf{a}_t$ output by the Controller at time $t$ to obtain the final prediction $\hat{X}_T$: $\hat{X}_T = \mathbf{a}_t \odot \hat{X}_T'$, where $\odot$ denotes the Hadamard product calculation. The predicted values of the nodes corresponding to elements with a value of 1 in the action will be locked, while the nodes corresponding to elements with a value of 0 will continue to update their predicted values in subsequent time steps.

## 4.3 CONTROLLER

We designed a reinforcement learning-based controller to determine whether each node has reached the optimal prediction time (Shao et al., 2023). It consists of a stop probability generation network and a policy network. In this paper, we use a fully connected network for the stop probability generation network and a Bernoulli-distribution-based policy network. First, the relative time information $t$ is replicated and expanded into a temporal feature vector $\mathbf{t} = \{t, t, \dots, t\} \in \mathbb{R}^n$, which is then concatenated with the output of the final unit of the decoder $\boldsymbol{H}_T \in \mathbb{R}^{n \times h}$ to form the input to the stop probability generation network: $\boldsymbol{C}_t \in \mathbb{R}^{n \times (h+1)}$. The calculation formula for the stop probability is as follows:

$$P_t = \sigma(\boldsymbol{W}_c \boldsymbol{C}_t + b_c) \tag{10}$$

where $\boldsymbol{W}_c$ and $b_c$ are the weights and biases of the fully connected network. The action $\mathbf{a}_t \in \mathbb{R}^n$ is obtained by performing a Bernoulli sampling according to the probability $P_t$. In practical

applications, we use the $\varepsilon$-greedy strategy to perform moderate exploration, preventing nodes from getting stuck in local optimal prediction time:

$$\mathbf{a}_t \sim Bernoulli((1 - \varepsilon)P_t) \tag{11}$$

where $\varepsilon$ represents the exploration probability, $a_t^i = 1$ means that time $t$ is the optimal prediction time for the node, and $a_t^i = 0$ means the opposite. The computation of the controller can be summarized as follows:

$$\mathbf{a}_t = \pi(\boldsymbol{C}_t) \tag{12}$$

where $\pi$ represents the entire policy of the controller.

## 5 EXPERIMENT

The datasets, baselines and specific parameter settings are presented in Appendix B.1. Experiments are conducted to compare the performance of the proposed DSTN against baselines, and the prediction timeliness of DSTN is demonstrated and analyzed. Furthermore, the impact of individual modules is illustrated through ablation experiments. Additionally, experimental results and analysis of the model hyper-parameters are provided in Appendix B.2.

### 5.1 PERFORMANCE COMPARISON

In Table 1, we compare the prediction performance of DSTN and baseline models under different data horizons. The **Data Horizon** refers to the proportion of data actually used for prediction relative to the complete observation window. For example, given a full observation window of $T = 12$, a 100% Data Horizon means that all available data are used as model input for prediction, whereas a 75% Data Horizon indicates that only the first 75% of the data—i.e., the first 9 time steps—are used as model input for prediction. This is equivalent to forcing the model to make predictions at different time points earlier than the end of the complete observation window. The purpose of compressing the data horizon is to explore the model's prediction performance under varying timeliness constraints.

It can be observed from Table 1 that: (1) DSTN demonstrates superior performance over all baseline models across the four datasets, indicating its excellent capability in diverse spatio-temporal prediction scenarios. Moreover, it exhibits strong robustness to various data characteristics, including graph size, periodic patterns, and temporal granularity. (2) DSTN outperforms two adaptive early spatio-temporal prediction models, STEMO and ESTGCN. This suggests that the proposed dynamic propagation matrix in DSTN can effectively filter irrelevant information while more accurately capturing early prediction information, thereby enhancing prediction accuracy. (3) Compared to other models, DSTN exhibits stronger robustness to the compression of the data horizon. For example, in the EMS dataset, as the data horizon decreases from 100% to 75%, 50%, and 25%, the MAE value of DSTN increases from 1.13 to 1.14, 1.15, and 1.17, with a growth percentage of around 1%, showing minimal impact. In contrast, the MAE value of STEMO increases from 1.29 to 1.32, 1.40, and 1.44, with the growth percentage reaching up to 6%, indicating a much greater impact than DSTN.

### 5.2 AVERAGE PREDICTION TIME

To evaluate the timeliness of DSTN, we present the **Average Prediction Time** of DSTN across various datasets and data horizons in Appendix Figure 4. The average prediction time is an important metric for early spatio-temporal prediction models, as it reflects the model's ability to balance prediction timeliness and accuracy and how much time can be saved for downstream tasks. The calculation of the average prediction time is as follows: $\bar{t}_{pre} = \frac{1}{n} \sum_{i=1}^{n} \frac{t_i^*}{T}$, where $\bar{t}_{pre}$ is the average prediction time, $T$ is the complete observation window, and $t_i^*$ is the optimal prediction time of node $v_i$ determined by the model.

From Figure 4, it can be observed that: (1) DSTN adaptively adjusts the optimal prediction time according to the characteristics of different datasets. The average prediction time of DSTN across the four datasets exhibit significant variations. For example, when the data horizon is 100%, the average prediction time for METR-LA, PEMS08, EMS and NYPD are 87.53%, 95.38%, 32.32% and 70.18%, respectively. (2) DSTN effectively achieves the objective of early prediction, thereby

Table 1: Performance comparison between DSTN and baseline models (averaged over five runs).

| Datasets | Data horizon | 100% | | 75% | | 50% | | 25% | |
|---|---|---|---|---|---|---|---|---|---|
| | | MAE | RMSE | MAE | RMSE | MAE | RMSE | MAE | RMSE |
| METR-LA | HA | 7.07 | 13.44 | 7.07 | 13.44 | 7.07 | 13.44 | 7.07 | 13.44 |
| | LSTM | 4.99 | 7.48 | 7.17 | 10.21 | 7.50 | 10.46 | 7.72 | 10.68 |
| | MTGNN | 2.40 | 4.10 | 3.04 | 5.67 | 3.31 | 6.67 | 3.65 | 7.79 |
| | DCRNN | 2.27 | **3.94** | 2.91 | 5.76 | 3.25 | 6.68 | 3.48 | 7.23 |
| | ASTGCN | 3.54 | 7.39 | 5.05 | 6.48 | 6.65 | 12.70 | 7.64 | 14.40 |
| | EARLIEST | 5.01 | 7.46 | 5.93 | 8.21 | 6.14 | 9.26 | 6.85 | 10.06 |
| | Graph-Wavenet | 2.38 | 4.21 | 3.22 | 6.45 | 3.81 | 7.70 | 4.35 | 8.65 |
| | STGCN | 2.68 | 4.47 | 3.35 | 6.02 | 3.61 | 6.68 | 4.18 | 7.82 |
| | ESTGCN | 2.81 | 5.46 | 3.07 | 5.96 | 3.33 | 6.48 | 3.46 | 7.66 |
| | STEMO | 2.75 | 4.75 | 3.39 | 6.22 | 3.98 | 7.36 | 4.34 | 8.18 |
| | **DSTN** | **2.18** | 4.23 | **2.48** | **5.46** | **2.94** | **6.35** | **3.23** | **7.02** |
| PEMS08 | HA | 36.08 | 54.84 | 36.08 | 54.84 | 36.08 | 54.84 | 36.08 | 54.84 |
| | LSTM | 19.98 | 32.92 | 21.28 | 34.57 | 25.07 | 37.67 | 30.54 | 45.10 |
| | MTGNN | 18.29 | 26.52 | 22.77 | 33.96 | 26.08 | 35.72 | 32.74 | 55.66 |
| | DCRNN | 16.53 | 24.95 | 20.35 | 30.77 | 24.16 | 36.15 | 30.10 | 43.39 |
| | ASTGCN | 18.22 | 26.99 | 23.86 | 33.85 | 27.21 | 38.74 | 31.69 | 46.32 |
| | EARLIEST | 17.59 | 24.90 | 21.76 | 33.60 | 24.64 | 36.95 | 29.31 | 43.97 |
| | Graph-Wavenet | 15.81 | 24.10 | 20.49 | 30.84 | 25.02 | 37.60 | 30.54 | 45.09 |
| | STGCN | 16.38 | **22.86** | 24.23 | 32.04 | 27.15 | 36.25 | 29.14 | 44.34 |
| | ESTGCN | 37.54 | 51.96 | 38.77 | 53.06 | 40.71 | 54.93 | 42.31 | 56.95 |
| | STEMO | 19.61 | 28.03 | 22.32 | 31.84 | 26.60 | 37.47 | 30.09 | 44.90 |
| | **DSTN** | **15.66** | 23.56 | **19.67** | **30.69** | **22.63** | **34.81** | **28.15** | **42.66** |
| EMS | HA | 1.49 | 2.53 | 1.49 | 2.53 | 1.49 | 2.53 | 1.49 | 2.53 |
| | LSTM | 1.21 | 2.25 | 1.26 | 2.36 | 1.30 | 2.45 | 1.33 | 2.48 |
| | MTGNN | 1.17 | 2.27 | 1.19 | 2.26 | 1.21 | 2.29 | 1.21 | 2.30 |
| | DCRNN | 1.21 | 2.29 | 1.29 | 2.28 | 1.38 | 2.34 | 1.55 | 2.58 |
| | ASTGCN | 1.24 | 2.31 | 1.28 | 2.42 | 1.33 | 2.49 | 1.38 | 2.55 |
| | EARLIEST | 1.22 | 2.32 | 1.25 | 2.41 | 1.26 | 2.39 | 1.31 | 2.50 |
| | Graph-Wavenet | 1.25 | 2.42 | 1.27 | 2.46 | 1.29 | 2.48 | 1.28 | 2.49 |
| | STGCN | 1.18 | 2.25 | 1.18 | 2.26 | 1.19 | 2.28 | 1.21 | 2.36 |
| | ESTGCN | 1.20 | 2.28 | 1.21 | 2.30 | 1.23 | 2.32 | 1.32 | 2.38 |
| | STEMO | 1.29 | 2.23 | 1.32 | 2.32 | 1.40 | 2.41 | 1.44 | 2.49 |
| | **DSTN** | **1.13** | **2.14** | **1.14** | **2.15** | **1.15** | **2.17** | **1.17** | **2.23** |
| NYPD | HA | 1.50 | 2.08 | 1.50 | 2.08 | 1.50 | 2.08 | 1.50 | 2.08 |
| | LSTM | 1.52 | 2.16 | 1.61 | 2.29 | 1.73 | 2.40 | 1.72 | 2.40 |
| | MTGNN | 1.48 | 2.09 | 1.52 | 2.16 | 1.54 | 2.20 | 1.56 | 2.23 |
| | DCRNN | 1.58 | 2.16 | 1.60 | 2.21 | 1.61 | 2.22 | 1.66 | 2.32 |
| | ASTGCN | 2.18 | 3.08 | 2.88 | 3.31 | 2.99 | 3.42 | 3.69 | 4.09 |
| | EARLIEST | 1.52 | 2.17 | 1.59 | 2.23 | 1.64 | 2.36 | 1.65 | 2.39 |
| | Graph-Wavenet | 1.48 | 2.05 | 1.50 | 2.08 | 1.54 | 2.14 | 1.58 | 2.18 |
| | STGCN | 1.45 | 2.05 | 1.48 | 2.08 | 1.49 | 2.09 | 1.51 | 2.16 |
| | ESTGCN | 1.49 | 2.06 | 1.51 | 2.10 | 1.52 | 2.11 | 1.53 | 2.13 |
| | STEMO | 1.46 | 2.05 | **1.46** | 2.06 | 1.49 | 2.10 | 1.50 | 2.12 |
| | **DSTN** | **1.45** | **1.98** | **1.46** | **1.99** | **1.47** | **1.99** | **1.48** | **2.02** |

providing additional preparation time for downstream tasks. For instance, with a 100% data horizon on the NYPD dataset, DSTN achieves an average prediction time of 70.18%, indicating that for any given node, DSTN can deliver accurate predictions approximately 14 hours in advance on average. (3) The prediction timeliness of DSTN is independent of the graph's topology and is primarily influenced by the complexity of the patterns and causality of the graph signals. The average prediction time does not increase as the graph size grows: the EMS dataset has more nodes than the NYPD dataset but has lower average prediction time. The more intricate the node correlations and the more dynamic the data patterns are, the longer DSTN requires to observe in order to generate reliable predictions.

Table 2: Ablation study results.

| Datasets | Data horizon | 100% | | 75% | | 50% | | 25% | |
|---|---|---|---|---|---|---|---|---|---|
| | | MAE | RMSE | MAE | RMSE | MAE | RMSE | MAE | RMSE |
| METR-LA | w/o $\boldsymbol{A}_t^{\mathcal{T}}$ | 2.48 | 4.74 | 3.17 | 6.53 | 3.61 | 7.49 | 4.09 | 8.38 |
| | w/o $\boldsymbol{A}^F$ | 2.79 | 5.59 | 3.37 | 6.87 | 3.85 | 7.91 | 4.12 | 8.49 |
| | w/o controller | 2.55 | 4.73 | 3.22 | 6.53 | 3.64 | 7.58 | 4.09 | 8.38 |
| | **DSTN** | **2.18** | **4.23** | **2.48** | **5.46** | **2.94** | **6.35** | **3.23** | **7.02** |
| PEMS08 | w/o $\boldsymbol{A}_t^{\mathcal{T}}$ | 15.73 | 23.93 | 20.14 | 31.03 | 23.51 | 36.08 | 29.28 | 43.47 |
| | w/o $\boldsymbol{A}^F$ | 17.19 | 26.12 | 21.08 | 31.73 | 24.02 | 36.29 | 30.15 | 45.46 |
| | w/o controller | 15.78 | 24.02 | 19.94 | 30.89 | 23.06 | 35.90 | 29.21 | 43.56 |
| | **DSTN** | **15.66** | **23.56** | **19.67** | **30.69** | **22.63** | **34.81** | **28.15** | **42.66** |
| EMS | w/o $\boldsymbol{A}_t^{\mathcal{T}}$ | 1.15 | 2.17 | 1.16 | 2.20 | 1.18 | 2.25 | 1.20 | 2.29 |
| | w/o $\boldsymbol{A}^F$ | 1.20 | 2.27 | 1.21 | 2.29 | 1.22 | 2.30 | 1.30 | 2.60 |
| | w/o controller | 1.14 | 2.15 | 1.18 | 2.16 | 1.20 | 2.19 | 1.23 | 2.28 |
| | **DSTN** | **1.13** | **2.14** | **1.14** | **2.15** | **1.15** | **2.17** | **1.17** | **2.23** |
| NYPD | w/o $\boldsymbol{A}_t^{\mathcal{T}}$ | 1.48 | 2.02 | 1.48 | 2.03 | 1.49 | 2.03 | 1.50 | 2.05 |
| | w/o $\boldsymbol{A}^F$ | 1.49 | 2.03 | 1.50 | 2.04 | 1.52 | 2.05 | 1.54 | 2.09 |
| | w/o controller | 1.47 | 2.00 | 1.48 | 2.02 | 1.50 | 2.03 | 1.52 | 2.07 |
| | **DSTN** | **1.45** | **1.98** | **1.46** | **1.99** | **1.47** | **1.99** | **1.48** | **2.02** |

## 5.3 ABLATION STUDY

To evaluate the impact of key components in the DSTN model, we conducted ablation experiments on all four datasets, with the results shown in Table 2. In the experiments, we removed three components from the DSTN model: w/o asynchronous spatio-temporal causal matrix (without $\boldsymbol{A}_t^{\mathcal{T}}$), w/o frequency similarity matrix (without $\boldsymbol{A}^F$), and w/o controller (without controller).

As shown in Table 2, the full DSTN model performs optimally across all data horizons. Specifically, when the asynchronous spatio-temporal causal matrix $\boldsymbol{A}_t^{\mathcal{T}}$ is removed, the model's performance decreases slightly, indicating that $\boldsymbol{A}_t^{\mathcal{T}}$ helps the model capture the causal relationships between short-term changes among nodes. When the frequency similarity matrix $\boldsymbol{A}^F$ is removed, there is a noticeable performance drop, suggesting that $\boldsymbol{A}^F$ plays a crucial role in identifying nodes with similar long-term patterns. Removing $\boldsymbol{A}_t^{\mathcal{T}}$ weakens the information provided by causal nodes, while removing $\boldsymbol{A}^F$ results in a complete loss of pattern information, leading to a more significant performance drop. When the controller component was removed, the performance of the model also exhibited a significant decline. This indicates that predictions made at time points autonomously determined by the controller are superior to those generated using the complete data horizon. This finding supports our argument that, in spatio-temporal prediction scenarios, a longer observation window does not necessarily lead to better predictive performance.

## 6 CONCLUSION

In this paper, we propose an adaptive early spatio-temporal prediction model with a dynamic propagation matrix (DSTN) to better balance the timeliness and accuracy of predictions. We compute a frequency similarity matrix to capture the similarity of long-term periodic patterns of signals and propose the Asynchronous Spatio-temporal Causal Frame Pair to capture the short-term causal relationships between different nodes. Then, a dynamic propagation matrix is constructed to filter out irrelevant information for early prediction.

Experimental results show that DSTN outperforms existing models on four real-world large-scale datasets. The experimental results indicate that DSTN tends to have a relatively large average prediction time for spatio-temporal data with complex causality. In the future, we will explore improvements in model architecture to further reduce prediction latency in complex datasets.

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

# A LOSS FUNCTION

## A.1 PREDICTION LOSS FUNCTION

We use the mean absolute error (MAE) between the predicted values and the true values as the prediction loss, which is defined as:

$$L_p = \frac{1}{n} \sum_{i=1}^{n} |\hat{X}_T^i - X_T^i| \tag{13}$$

where $\hat{X}_T^i$ is the predicted value, $X_T^i$ is the true value, and n is the number of nodes.

## A.2 CONTROLLER LOSS FUNCTION

The purpose of adding the controller loss is to enable the controller to more accurately determine the optimal prediction time. Based on the final prediction results, the corresponding reward $\mathbf{r} \in \mathbb{R}^n$ is calculated. For each node, if the absolute error between the predicted result and the true value is within a certain range $\epsilon$, the reward is 1; otherwise, the reward is -1. The specific calculation can be expressed as:

$$r_i = \begin{cases} 1, & |\hat{X}_T^i - X_T^i| \leq \epsilon \\ -1, & |\hat{X}_T^i - X_T^i| > \epsilon \end{cases} \tag{14}$$

Our goal is to maximize the probability of actions that lead to positive rewards by constructing a loss function. Therefore, the objective can be expressed as:

$$L_c = -\mathbb{E}[\mathbf{r} \sum_{t=0}^{t^*} \log \pi(\mathbf{a}_t | C_t)] \tag{15}$$

where $\log \pi(\mathbf{a}_t | C_t)$ represents the probability of taking the action $\mathbf{a}_t$ given that the input to the controller is $C_t$. However, the variance in Equation (15) can be quite large. To reduce the variance and accelerate the convergence of the controller, following the approach in (Mnih et al., 2014), we introduce a baseline network to adjust the reward. The adjusted controller loss function is as follows:

$$L_c = -\mathbb{E}[\sum_{t=0}^{t^*} \log \pi(\mathbf{a}_t | C_t)(\mathbf{r} - \mathbf{b}_t)] \tag{16}$$

where $\mathbf{b}_t$ is the correction term from the baseline network. In this paper, a fully connected network is used as the baseline network, with its input being $C_t$. The baseline network is trained by minimizing the mean squared error (MSE) between $\mathbf{b}_t$ and $\mathbf{r}$.

The final loss function for the model is expressed as:

$$L = L_p + L_b + L_c \tag{17}$$

where $L_b$ is the loss for the baseline network.

# B EXPERIMENT

## B.1 EXPERIMENT SETTINGS

### B.1.1 DATASETS

We conduct experiments on four real-world public large-scale datasets varying in type, graph size, geographical distribution, temporal granularity, and periodic patterns, which are widely used to test the performance of spatio-temporal prediction models (Wu et al., 2019; Shao et al., 2024; 2023):

METR-LA (Jagadish et al., 2014): The dataset contains public transportation speed data from 207 sensors on the Los Angeles Expressway. The data is sampled every 5 minutes and spans the period from March 1, 2012, to June 30, 2012.

PEMS08[1]: The dataset contains traffic flow data in San Bernardino, collected by 170 detectors on 8 roads. The data is sampled every 5 minutes and spans the period from July to August in 2016.

EMS[2]: The dataset contains emergency data from the New York Fire Department, including 145 areas based on postal code. The data is sampled every 1 hour and spans the period from January 1, 2011 to November 30, 2011.

NYPD[3]: The dataset is a collection of crime data from the New York Police Department. The dataset is divided into 77 regions based on administrative regions. The data is sampled every 4 hours and spans the period from January 1, 2014 to December 31, 2015.

Table 3: Statistics of datasets

| Datasets | Samples | Nodes | Sample rate | Domain |
|---|---|---|---|---|
| METR-LA | 34272 | 207 | 5 min | Traffic Speed |
| PEMS08 | 17856 | 170 | 5 min | Traffic Flow |
| EMS | 7992 | 145 | 1 h | Emergency |
| NYPD | 4386 | 77 | 4 h | Crime |

Detailed statistics of the datasets are presented in Table 3. For all datasets, 70% of the data is allocated for training, 20% for testing, and the remaining 10% for validation. The experiments are conducted on a Windows system equipped with an Nvidia GeForce RTX 3080 GPU and Intel(R) Core(TM) i7-10700F CPU @ 2.90GHz.

### B.1.2 HYPER-PARAMETERS SETTINGS

In the experiments, we set the complete observation window $T = 12$, the ratio coefficient $\alpha = 0.01$, the GCRU hidden state dimension $h = 16$, the error range $\epsilon = 0.5$, and the exploration probability of the controller $\varepsilon = 0.1$. The model was trained with a batch size of 64 and the Adam optimizer.

### B.1.3 BASELINES AND METRICS

We compare the proposed DSTN with the following baselines for spatio-temporal prediction and early spatio-temporal prediction tasks: HA (Liu & Guan, 2004), LSTM (Hochreiter, 1997), MTGNN (Wu et al., 2020), DCRNN (Li et al., 2017), ASTGCN (Guo et al., 2019), EARLI-EST (Hartvigsen et al., 2019), Graph-WaveNet (Wu et al., 2019), STGCN (Yu et al., 2017), EST-GCN (Shao et al., 2023), STEMO (Shao et al., 2024).

We use MAE (Mean Absolute Error) and RMSE (Root Mean Squared Error) to evaluate the performance of all models, which are two common metrics used to measure the average prediction error in regression tasks.

### B.2 HYPER-PARAMETERS STUDY

Through systematic experimentation, we investigated the impact of three key hyper-parameters on model performance with dataset PEMS08: the ratio coefficient $\alpha$, exploration probability $\varepsilon$, and error range $\epsilon$. The results are presented in Figure 5.

Specifically, we evaluated model performance across varying values of $\alpha$ (0.01, 0.05, 0.1, 0.5, 1) and $\varepsilon$ (0.1, 0.15, 0.2, 0.25, 0.3). The results shown in Figure 5a and Figure 5b demonstrate that variations in parameters $\alpha$ and $\varepsilon$ have no significant impact on model performance. The ratio coefficient $\alpha$ serves to adjust the weights of asynchronous spatio-temporal causal frame pairs in the ASCM with varying similarity levels. DSTN leverages these weights to assess the strength of causal relationships between nodes. Experimental results demonstrate that DSTN effectively identifies different levels of causal relationships across a wide $\alpha$ value range, with no significant impact on final

---

[1] https://pems.dot.ca.gov/

[2] https://data.cityofnewyork.us/Public-Safety/EMS-Incident-Dispatch-Data/76xm-jjuj

[3] https://www.kaggle.com/datasets/mrmorj/new-york-city-police-crime-data-historic

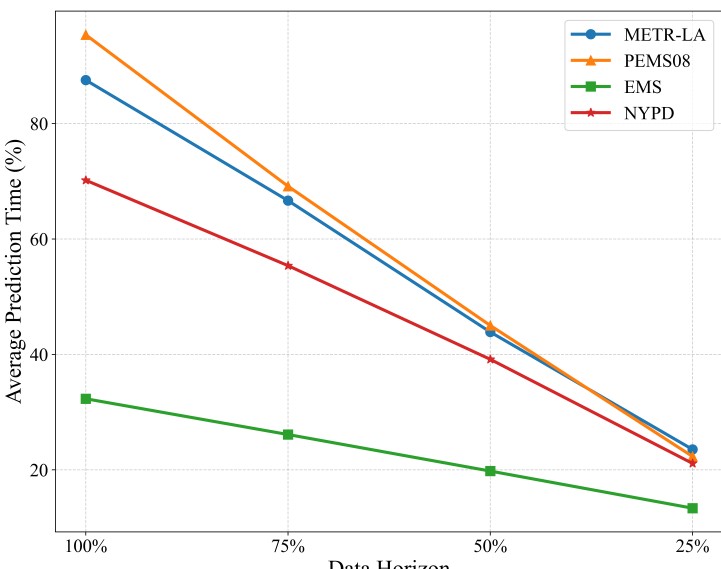

Figure 4: The average prediction time of the DSTN under different data horizon in the METR-LA, PEMS08, EMS, and NYPD datasets. It can be observed that different datasets, depending on their own data characteristics, exhibit different average prediction time under the same data horizon.

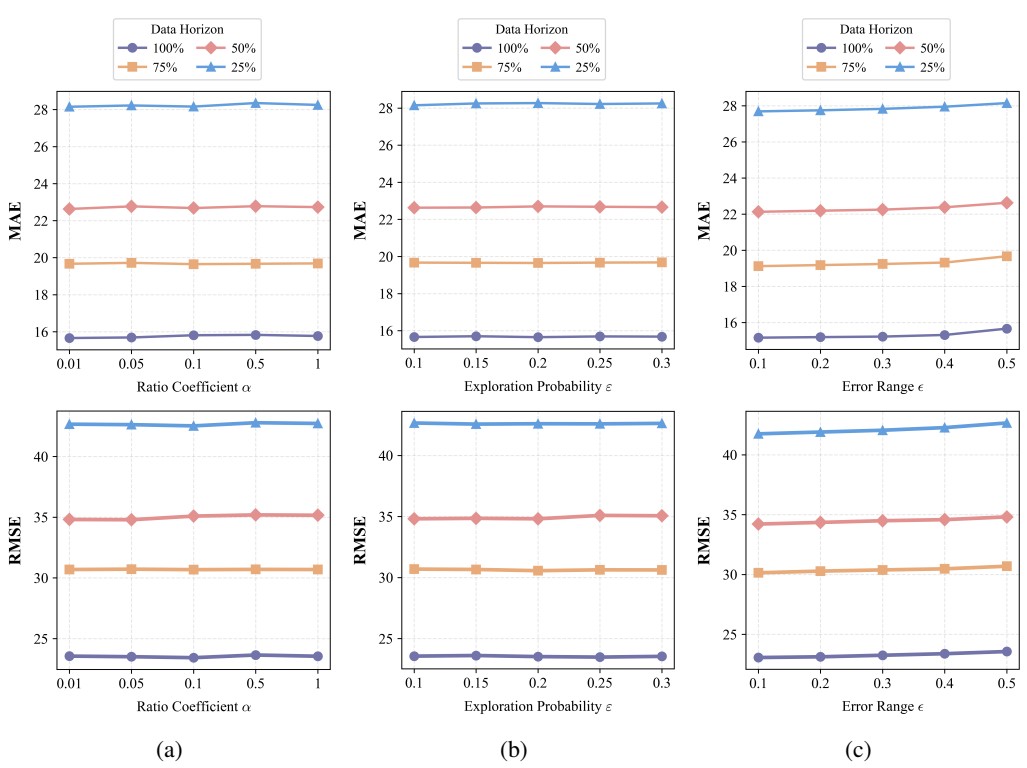

Figure 5: Impact of ratio coefficient $\alpha$, exploration probability $\varepsilon$ and error range $\epsilon$ on model performance.

model performance. The exploration probability $\varepsilon$ is designed to prevent the controller module from converging to local optima during training, which could impair its ability to accurately determine the optimal prediction time. Our investigation of multiple exploration probability values similarly re-

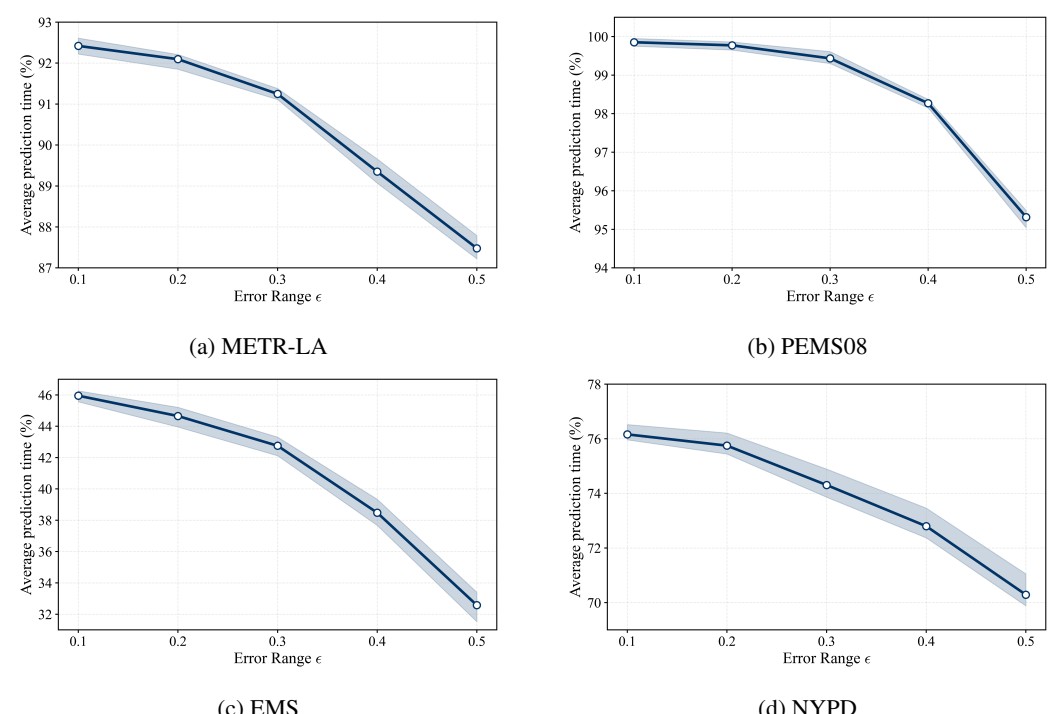

Figure 6: Impact of error range $\epsilon$ on average prediction time at 100% Data Horizon. The line represents the mean values from multiple experiments, while the shaded area indicates the range between the maximum and minimum values observed across these experiments.

Table 4: Impact of ratio coefficient $\alpha$ and exploration probability $\varepsilon$ on average prediction time at 100% Data Horizon.

| Dataset | $\alpha = 0.01$ | $\alpha = 0.05$ | $\alpha = 0.1$ | $\alpha = 0.5$ | $\alpha = 1$ |
|---|---|---|---|---|---|
| METR-LA | 87.48%±0.15% | 87.81%±0.18% | 87.22%±0.21% | 86.95%±0.17% | 88.10%±0.20% |
| PEMS08 | 95.35%±0.12% | 95.12%±0.20% | 95.24%±0.15% | 95.27%±0.19% | 95.15%±0.16% |
| EMS | 32.21%±0.25% | 32.40%±0.24% | 32.25%±0.28% | 32.35%±0.20% | 32.30%±0.26% |
| NYPD | 70.16%±0.18% | 70.25%±0.16% | 70.10%±0.15% | 70.20%±0.18% | 70.18%±0.19% |
| Dataset | $\varepsilon = 0.1$ | $\varepsilon = 0.15$ | $\varepsilon = 0.2$ | $\varepsilon = 0.25$ | $\varepsilon = 0.3$ |
| METR-LA | 87.55%±0.11% | 87.60%±0.13% | 87.52%±0.12% | 87.58%±0.19% | 87.53%±0.15% |
| PEMS08 | 95.38%±0.17% | 95.42%±0.16% | 95.35%±0.17% | 95.40%±0.20% | 95.37%±0.18% |
| EMS | 32.32%±0.22% | 32.28%±0.25% | 32.35%±0.23% | 32.33%±0.26% | 32.38%±0.22% |
| NYPD | 70.18%±0.13% | 70.22%±0.19% | 70.15%±0.18% | 70.23%±0.16% | 70.17%±0.20% |

veals that its setting within reasonable bounds shows no statistically significant effect on the model's performance.

Compared to the ratio coefficient and exploration probability, the error range $\epsilon$ demonstrates a noticeable impact on model performance, manifesting in both prediction accuracy and timeliness. As shown in Figure 5c, as the error range increases from 0.1 to 0.5, the model's MAE and RMSE exhibit a slight increase, while the average prediction time demonstrates an exponential decay ($e^{-x}$), as illustrated in Figure 6b. To further verify the impact of the error range on the average prediction time, we also conducted experiments on the remaining datasets. The results, presented in Figures 6a, 6c, and 6d, demonstrate a consistent decreasing trend in the average prediction time across all datasets as the error range increases. This is because the error range serves to construct the loss function of the controller module, guiding the controller to seek the optimal prediction time. When the error range is larger, the controller does not attempt to further increase prediction time to improve accuracy, as low prediction accuracy can evade penalties. Meanwhile, experimental results indicate

that the ratio coefficient $\alpha$ and exploration probability $\varepsilon$ have no significant impact on the average prediction time. Detailed results are presented in Table 4.

## C  THE USE OF LARGE LANGUAGE MODELS

In this paper, large language models (LLMs) were employed to polish and refine parts of the manuscript. The LLM was used solely for linguistic enhancement and did not contribute to the conceptual development, technical content, or analytical findings of the research.

