# OpenReview forum: "DSTN: Early Spatio-Temporal Forecasting with Dynamic Propagation"
_ICLR.cc/2026/Conference — ICLR 2026 Conference Withdrawn Submission_

### Official Review · Reviewer_hfCo · 2025-10-17

**Soundness:** 2
**Presentation:** 2
**Contribution:** 2
**Rating:** 4
**Confidence:** 4

**Summary:**

This paper proposes DSTN (Dynamic Spatio-Temporal Network), an adaptive early spatio-temporal prediction model that balances prediction timeliness and accuracy. The core contribution lies in constructing a dynamic propagation matrix by combining frequency-domain similarity (capturing long-term periodic patterns) and asynchronous spatio-temporal causal relationships (capturing short-term dynamic causality). A reinforcement learning-based controller determines the optimal prediction time for each node. The authors evaluate DSTN on four real-world datasets (METR-LA, PEMS08, EMS, NYPD) and claim superior performance over baselines in early prediction scenarios.

**Strengths:**

**S1. Important and well-motivated problem**: Early spatio-temporal prediction is practically valuable for time-sensitive applications (crime prevention, traffic management). The paper clearly articulates the motivation.

**S2. Novel conceptual framing**: The "Asynchronous Spatio-temporal Causal Frame Pair" concept, offers an interesting perspective on time-delayed causality that differs from sliding window approaches.

**S3. Comprehensive experiments**: The paper tests on four datasets with multiple baselines and includes ablation studies, demonstrating effort in empirical evaluation.

**Weaknesses:**

**W1. Weak theoretical foundation:** Phase-based delay estimation (Eq. 4) assumes **strict periodicity**, which fails for non-stationary signals with multiple frequency components and time-varying patterns. And there’s No analysis of estimation error propagation to downstream causal matrix computation.

**W2. Marginal and inconsistent performance gains:** RMSE on METR-LA/PEMS08 is not SOTA. Improvements over STEMO/ESTGCN are incremental.

**W3. Unjustified design choices**: There’s no justification about the method. For example, why construct dynamic propagation matrix by simply adding $A^F$ and ASCM with different scales(Eq. 8)?

**W4. Oversimplified controller**: Single-layer FC network may be insufficient for complex stopping decisions. There’s no proof on the greedy strategy. And RL reward functions are standard.

**W5. Questionable "large-scale" claim**: The authors claimed that they evaluated on four “large-scale dataset”. But to the best of my knowledge, LargeST[1] has 8,600+ nodes and 500,000+ frames, which is much larger than the dataset used in the manuscript.

**W6. Lack of Efficiency Study and Complexity Discussion.**

[1] Liu X, Xia Y, Liang Y, et al. Largest: A benchmark dataset for large-scale traffic forecasting[J]. Advances in Neural Information Processing Systems, 2023, 36: 75354-75371.

**Questions:**

Q1. How does the method handle nodes with multiple dominant frequencies or no clear dominant frequency? Does it select the strongest frequency or use all frequencies?

Q2. In Eq. (8), $A^F$ and ASCM have different dimensions (semantically and possibly numerically). Why is simple addition justified? This implies equal contributions, which seems arbitrary. Additionally, ASCM varies with $t$, shouldn't temporal decay be applied to earlier timesteps? Otherwise, t=0 and t=11 contribute equally to prediction at t=12, which is unreasonable.

Q3. Causality validation: Can you provide evidence that DTW-based similarity truly captures causality (not just correlation)? For example, synthetic experiments with known ground-truth causal graphs?

Q4. Computational cost: What is the runtime complexity of computing A_D at each timestep? How does it compare to baselines?

---

> ### Author Response · Authors · 2025-11-20
>
> **Q1: How does the method handle nodes with multiple dominant frequencies...**
>
> R1: Thank you for your question. First, to answer your question: whether there are multiple dominant frequencies or no clear dominant frequency, when calculating the $A^F$, all frequency components are considered, but when calculating the time delay, only the strongest frequency component (the dominant frequency) is focused on.
>
> The logic of our method is as follows:
> 1. For node pairs with consistent dominant frequencies, similar periodic variations, and strong short-term dynamic causality, they will simultaneously receive high weights in both the $A^F$ and the ASCM.
> 2. For node pairs with inconsistent dominant frequencies but similar periodic variations and strong short-term dynamic causality, they will receive high weights in the $A^F$ but lose the high weights in the ASCM (because the time delay cannot be calculated due to the inconsistent dominant frequencies).
> 3. For node pairs with consistent dominant frequencies and similar periodic variations but no short-term dynamic causality, they will receive high weights in the $A^F$ but lose the high weights in the ASCM (due to the large DTW distance between the asynchronous spatio-temporal causal frame pairs).
>
> Thus, it can be seen that if two nodes with similar periodic variations have signals with multiple dominant frequencies or no clear dominant frequency, it is the best scenario if their dominant frequencies are consistent. If their dominant frequencies are inconsistent, causing the potential loss of high weights in the ASCM, the high weights in the $A^F$ can still support the correlation between them, preventing it from being completely lost. This is also the reason why we use two matrices, $A^F$ and ASCM, to construct the dynamic propagation matrix.
>
> **Q2: Why is simple addition justified and temporal decay be applied to earlier timesteps...**
>
> R2: In our experiments, we tried setting different weights. The results showed that this direct addition method is the most universal and stable approach. Furthermore, a similar operation of directly adding to construct the propagation matrix exists in STEMO [1], an early prediction model analogous to ours. Therefore, we retained this method.
>
> **Q3: Questionable "large-scale" claim...**
>
> R3: The four datasets selected in this paper are recognized in the fields of traditional spatio-temporal prediction and early spatio-temporal prediction as benchmarks that can effectively evaluate model performance [1, 2, 3, 4]. Compared to other similar datasets, these are considered relatively large-scale, and this terminology has been similarly used in other publications [1, 2]. We had no intention to exaggerate.
>
> **Q4: Causality validation: Can you provide evidence that DTW-based similarity truly captures causality...**
>
> R4: We agree that using DTW distance alone is insufficient for accurately measuring causality. However, our approach involves first estimating the time delay between node signals through frequency-domain analysis. We then use this estimated delay to identify a prior signal frame that is temporally aligned in a causally relevant manner with the current (posterior) frame. If the DTW distance between these specifically aligned frames is extremely small, it provides strong evidence for a significant causal relationship between the two signals concerning that particular dynamic change. Conducting experimental validation on synthetic data with known ground-truth causal graphs to demonstrate the effectiveness of ASCM is an excellent suggestion. Thank you for this recommendation, we will explore this approach in our subsequent work.
>
> **Q5: Oversimplified controller**
>
> R5: There is currently no evidence indicating that the existing controller model cannot handle complex stopping decisions. The controller component is not the main contribution of this paper; therefore, for both the greedy strategy and the RL reward functions, we employed commonly used and stable solutions.
>
> [1] Shao W, Kang Y, Peng Z, et al. Stemo: Early spatio-temporal forecasting with multi-objective reinforcement learning[C]//Proceedings of the 30th ACM SIGKDD Conference on Knowledge Discovery and Data Mining. 2024: 2618-2627.
>
> [2] Shao W, Peng Z, Kang Y, et al. Early spatiotemporal event prediction via adaptive controller and spatiotemporal embedding[C]//2023 IEEE International Conference on Data Mining (ICDM). IEEE, 2023: 1307-1312.
>
> [3] Kong W, Guo Z, Liu Y. Spatio-temporal pivotal graph neural networks for traffic flow forecasting[C]//Proceedings of the AAAI conference on artificial intelligence. 2024, 38(8): 8627-8635.
>
> [4] Dong Z, Jiang R, Gao H, et al. Heterogeneity-informed meta-parameter learning for spatiotemporal time series forecasting[C]//Proceedings of the 30th ACM SIGKDD conference on knowledge discovery and data mining. 2024: 631-641.

---

> > ### Comment · Reviewer_hfCo · 2025-11-27
> >
> > Thanks to the authors' detailed responses. While I appreciate the clarifications provided, several of my core concerns remain inadequately addressed. For example, in Answer 2, you claim experimental validation by setting different weights was conducted, but no result data were presented to substantiate these claims. Mentioned questions (Q4) regarding computational cost were not directly addressed in your rebuttal. I encourage you to incorporate these elements in future revisions of this work. In the end, I will maintain my score.

---

### Official Review · Reviewer_1R18 · 2025-10-25

**Soundness:** 2
**Presentation:** 2
**Contribution:** 3
**Rating:** 2
**Confidence:** 4

**Summary:**

This paper proposes an adaptive early spatio-temporal forecasting model designed to deliver reliable predictions before the full observation window is available. It leverages long-term pattern similarity to support early forecasting, introduces an Asynchronous Spatio-temporal Causal Frame Pair mechanism to estimate pairwise time delays through causal relationships, and constructs a dynamic propagation matrix to filter out irrelevant signals. Experiments on four datasets demonstrate consistent improvements over baseline methods.

**Strengths:**

1. The paper's study on timeliness is valuable and has clear applications to crime prediction and traffic congestion forecasting.

2. Modeling inter-node time delays using dominant frequency and phase shift is also an interesting and well-motivated approach.

3. Using a reinforcement-learning controller to determine whether each node has reached its optimal prediction time helps avoid misleading signals.

**Weaknesses:**

1. The time delay is estimated, but only one example is provided to show that the estimated value is close to the actual delay. A more thorough analysis of time-delay estimation accuracy would strengthen the paper. Since time delays likely vary across node pairs, it would be helpful to report accuracy metrics across multiple pairs and examine how estimation errors affect prediction performance or propagate through the model.

2. The experimental setup is not clearly described. Since the optimal prediction time may differ across nodes, the controller’s decision process for determining this per-node timing is unclear. It would be helpful to clarify the state how the model handles per-node heterogeneity.

3. The target prediction horizon is not clearly stated. The paper mentions an observation window of t=12, but it is unclear what the corresponding prediction window is. Additionally, the average stopping time per dataset is not reported, which is important for understanding the model’s early prediction behavior.

4. The data‑horizon notation is confusing. The use of 100%, 75%, 50%, and 25% would be clearer if mapped explicitly to observation lengths, such as t=12,9,6,3, along with an explanation of how these settings reflect early‑prediction behavior. Although the experiments show that the model performs well with limited observations, it remains unclear how the method determines the optimal prediction time under a fixed observation length.

5. Although the paper motivates crime and congestion forecasting, there are no experiments on crime or traffic-congestion datasets, which weakens the application claims.

6. The paper lacks case studies or visualization showing that the method truly filters unnecessary signals and adapts prediction times effectively.

**Questions:**

1. The time delay is estimated, but only one example is provided to show that the estimated value is close to the actual delay. A more thorough analysis of time-delay estimation accuracy would strengthen the paper. Since time delays likely vary across node pairs, it would be helpful to report accuracy metrics across multiple pairs and examine how estimation errors affect prediction performance or propagate through the model.

2. The experimental setup is not clearly described. Since the optimal prediction time may differ across nodes, the controller’s decision process for determining this per-node timing is unclear. It would be helpful to clarify the state how the model handles per-node heterogeneity.

3. The target prediction horizon is not clearly stated. The paper mentions an observation window of t=12, but it is unclear what the corresponding prediction window is. Additionally, the average stopping time per dataset is not reported, which is important for understanding the model’s early prediction behavior.

4. The data‑horizon notation is confusing. The use of 100%, 75%, 50%, and 25% would be clearer if mapped explicitly to observation lengths, such as t=12,9,6,3, along with an explanation of how these settings reflect early‑prediction behavior. Although the experiments show that the model performs well with limited observations, it remains unclear how the method determines the optimal prediction time under a fixed observation length.

5. Although the paper motivates crime and congestion forecasting, there are no experiments on crime or traffic-congestion datasets, which weakens the application claims.

6. The paper lacks case studies or visualization showing that the method truly filters unnecessary signals and adapts prediction times effectively.

---

> ### Author Response · Authors · 2025-11-20
>
> **Q1: The time delay is estimated, but only one example is provided...**
>
> R1: We appreciate the suggestion. In the subsequent version, we will use more examples to demonstrate the accuracy of the time delay estimation based on frequency-domain analysis and explore the impact of time delay error on prediction performance.
>
> **Q2: The experimental setup is not clearly described...**
>
> R2: If a node can obtain more early prediction information, the controller will determine an earlier prediction time for that node. Early prediction information includes two aspects: the periodic information of the node's own signal, and the prediction information provided by causal nodes. In this paper, we primarily aimed to demonstrate the overall improvement in prediction timeliness achieved by DSTN, while the timeliness analysis for specific nodes was omitted. We will add detailed case studies later to more comprehensively illustrate the role of the controller. Thank you for the suggestion.
>
> **Q3: The target prediction horizon is not clearly stated...**
>
> R3: Detailed dataset information is provided in **Appendix B.1.1**. The average stopping time (**average prediction time**) for each dataset are analyzed in detail in **Section 5.2** and visualized in **Appendix Figure 4**. The observation window length t=12 mentioned in the paper refers to 12 time steps, where the duration of a time step is equivalent to the sampling rate of the dataset. This study uses four datasets: METR-LA, PEMS08, EMS, and NYPD, with sampling rates of 5min, 5min, 1h, and 4h respectively. Therefore, their corresponding length of observation windows are 1h, 1h, 12h, and 48h (**12 * sampling rate**).
>
> **Q4: The data horizon notation is confusing...**
>
> R4: Although the observation window length is set to t=12 in this paper and 100%, 75%, 50%, and 25% data horizons indeed correspond to prediction time steps of t=12, 9, 6, and 3 respectively, this length can be set to different values according to practical situations. Therefore, we believe that using percentages to denote the data horizon is a more universal method than using specific numerical values. The explanation of how these settings reflect early prediction behavior is in **Section 5.1**. Here is some additional clarification:
> 1.  Since traditional spatio-temporal prediction models lack inherent early prediction capability, forcing these models to make predictions at different data horizons effectively turns them into fixed early prediction models, making the comparison with our proposed early prediction model DSTN more reasonable.
> 2.  The results under different data horizons more intuitively demonstrate that DSTN can achieve better prediction results using shorter observation time. For example, DSTN's prediction accuracy using a 75% data horizon surpasses the prediction accuracy of baselines using a 100% data horizon, as can be easily observed in **Table 1**.
> 3.  To obtain the average prediction time of DSTN under different data horizons to illustrate the role of the controller, visualized in **Appendix Figure 4**.
> 4.  The results show that even when traditional spatio-temporal prediction models are forced to compress their data horizon to near DSTN's average prediction time, they cannot achieve results comparable to DSTN. For instance, on the NYPD dataset, DSTN's average prediction time at the 100% data horizon is 70.18%. However, at the 75% data horizon, no baseline's prediction accuracy exceeds DSTN's performance at the 100% data horizon. This directly proves the necessity and effectiveness of adaptive early prediction and the controller.
>
>
> Regarding how our method determines the optimal prediction time under a fixed observation length, the approach is described in **Section 4.3**. Here we provide some additional explanation: Assume the fixed observation length is $T$. For a specific node $k$, starting from time step $t=0$, the spatio-temporal encoder-decoder module generates a prediction for the target time step based on the correlations and causality between nodes. If the controller deems the current conditions insufficient for this node to make an accurate prediction at this moment, it discards the prediction and waits for the next time step to repeat the process. This continues until the controller determines that the conditions are sufficient for the node to make an accurate prediction. That specific time point is identified as the optimal prediction time $t_k^{*}<T$ in the current prediction instance.
>
> **Q5: Although the paper motivates crime and congestion forecasting...**
>
> R5: Detailed information of the used datasets is provided in **Appendix B.1.1**, where **PEMS08** is a **traffic flow** dataset and **NYPD** is a **crime** dataset.

---

### Official Review · Reviewer_VtYE · 2025-10-31

**Soundness:** 3
**Presentation:** 2
**Contribution:** 3
**Rating:** 4
**Confidence:** 4

**Summary:**

This paper proposes an adaptive early spatio-temporal prediction model with a Dynamic Spreading Transmission Network (DSTN), aimed at addressing the trade-off between timeliness and accuracy in spatio-temporal forecasting tasks. DSTN captures causal dependencies between nodes by constructing a dynamic propagation matrix, identifies long-term patterns using frequency-domain similarity, and captures short-term causal relationships through asynchronous spatio-temporal causal frame pairs. Additionally, a reinforcement learning-based controller is employed to adaptively determine the optimal prediction time for each node. Experiments demonstrate that DSTN outperforms existing baseline methods on four large-scale real-world datasets.

**Strengths:**

1. The study of early spatio-temporal prediction is both interesting and practically meaningful.
2. The proposed model achieves superior performance compared to baseline models in early prediction tasks.

**Weaknesses:**

1. The paper lacks comparisons with a wide range of SOTA spatio-temporal baselines. Most of the general ST baselines used for comparison were proposed before 2020, which weakens the persuasiveness of the experimental results.
2. The early prediction task seems to be a subset of long-term forecasting, e.g., predicting data from 12:00 to 18:00 based on data from 00:00 to 06:00 is included in predicting from 06:00 to 18:00. Has the author explored or experimented with this dimension?
3. The paper equates a small DTW (Dynamic Time Warping) distance with strong causality, but correlation does not necessarily imply causation.

**Questions:**

Please refer to Weaknesses

---

> ### Author Response · Authors · 2025-11-20
>
> **Q1: The paper lacks comparisons with a wide range of SOTA spatio-temporal baselines...**
>
> R1: Since our work focuses on early spatio-temporal prediction, the baselines in our experiments include two of the most recent early spatio-temporal prediction models (**STEMO 2024** and **ESTGCN 2023**), and do not include newer traditional spatio-temporal prediction models.
>
> **Q2: The early prediction task seems to be a subset of long-term forecasting...**
>
> R2: You are absolutely correct. If the prediction timeliness of an early spatio-temporal model for a specific period aligns with the scenario in your example, then it can indeed be viewed as a form of long-term forecasting during that period. However, **we must clarify that early spatio-temporal prediction and long-term forecasting are not directly equivalent concepts**. The improvement in prediction timeliness offered by early prediction is defined relative to traditional spatio-temporal prediction (which produces forecasts only after observing a fixed-length observation window). Our model aims to provide equally or more accurate predictions without waiting for the entire fixed observation window to elapse. Furthermore, the optimal prediction time point can vary dynamically for each node based on its current state. That said, under conditions of good timeliness, the prediction pattern for some nodes can indeed be considered a type of long-term forecasting, and we acknowledge that these perspectives are not mutually exclusive.
>
> **Q3: The paper equates a small DTW...**
>
> R3: We agree that using DTW distance alone is insufficient for accurately measuring causality. However, our approach involves first estimating the time delay between node signals through frequency-domain analysis. We then use this estimated delay to identify a prior signal frame that is temporally aligned in a causally relevant manner with the current (posterior) frame. If the DTW distance between these specifically aligned frames is extremely small, it provides strong evidence for a significant causal relationship between the two signals concerning that particular dynamic change.

---

> > ### Comment · Reviewer_VtYE · 2025-11-22
> >
> > Thank you very much for the author's response. Some of my concerns have been addressed. However, I still have a few concerns, such as the performance gap and advantages between early prediction and long-term forecasting using general spatio-temporal models. Therefore, I have decided to maintain my original score.

---

### Official Review · Reviewer_d3oM · 2025-10-31

**Soundness:** 2
**Presentation:** 3
**Contribution:** 2
**Rating:** 4
**Confidence:** 5

**Summary:**

This paper aims to balance between prediction timeliness and accuracy is essential for spatio-temporal forecasting. The adaptive early spatio-temporal prediction model with a dynamic propagation matrix is proposed, which captures causal relationships between nodes to enhance prediction timeliness while maintaining accuracy. Experimental results on four real-world datasets demonstrate that the performance of the proposed model generally outperforms all baselines.

**Strengths:**

- This paper presents an adaptive early spatio-temporal prediction model with a dynamic propagation matrix, which is a trade-off solution between the timeliness and accuracy of spatio-temporal predictions.

- The proposed method is simple yet effective according to the experimental results.

**Weaknesses:**

- There is no evidence that the long-term characteristic is ignored in existing studies [1].

- Many recent works focused on the causality of spatio-temporal data [2, 3].

- The baseline methods used in the evaluation are relatively outdated. It would be better to compare the latest state-of-the-art methods, such as HimNet [4] and STPGNN [5].

[1] Foundation models for spatio-temporal data science: A tutorial and survey[C]//Proceedings of the 31st ACM SIGKDD Conference on Knowledge Discovery and Data Mining V. 2. 2025: 6063-6073.

[2] Nuwadynamics: Discovering and updating in causal spatio-temporal modeling[C]//The Twelfth International Conference on Learning Representations. 2024.

[3] Ma J, Cui Z, Wang B, et al. Causal learning meet covariates: Empowering lightweight and effective nationwide air quality forecasting. IJCAI 2025.

[4] Dong Z, Jiang R, Gao H, et al. Heterogeneity-informed meta-parameter learning for spatiotemporal time series forecasting[C]//Proceedings of the 30th ACM SIGKDD conference on knowledge discovery and data mining. 2024: 631-641.

[5] Spatio-temporal pivotal graph neural networks for traffic flow forecasting[C]//Proceedings of the AAAI conference on artificial intelligence. 2024, 38(8): 8627-8635.

**Questions:**

1. You mention that existing studies overlook long-term temporal dependencies in the Introduction section. Could you please provide empirical evidence to support this observation? How does your method explicitly capture long-term dynamics compared to prior models?

2. Several recent works have investigated causal spatio-temporal modeling. Could you elaborate on how your model differs from existing causal-based frameworks?

3. The baseline methods in the current experiments appear relatively dated. Is it possible to compare with more recent state-of-the-art models?

4. Could you provide insight into the computational complexity of the proposed method?

---

> ### Author Response · Authors · 2025-11-20
>
> **Q1: Long-term characteristic and causality.**
>
> R1: The point we intended to make is that, within the specific field of early spatio-temporal prediction, no existing work has attempted to utilize long-term characteristics and causality to achieve early prediction, as illustrated in **Introduction Page 2**. We are fully aware that related work exists in traditional spatio-temporal prediction; in fact, our inspiration was drawn precisely from those studies. However, the absence of focus on long-term characteristics and causality in early spatio-temporal prediction is, in our view, quite unreasonable. We believe that the key to achieving accurate early prediction lies precisely in capturing the correlations and causality between nodes.
>
> **Q2: The baseline methods in the current experiments appear relatively dated.**
>
> R2: Since our work focuses on early spatio-temporal prediction, the baselines in our experiments include two of the most recent early spatio-temporal prediction models (**STEMO 2024** and **ESTGCN 2023**), and do not include newer traditional spatio-temporal prediction models.

---

### Note · Authors · 2025-11-29

I have read and agree with the venue's withdrawal policy on behalf of myself and my co-authors.